# Soybean Seedling Root Segmentation Using Improved U-Net Network

**DOI:** 10.3390/s22228904

**Published:** 2022-11-17

**Authors:** Xiuying Xu, Jinkai Qiu, Wei Zhang, Zheng Zhou, Ye Kang

**Affiliations:** 1College of Engineering, Heilongjiang Bayi Agricultural University, Daqing 163319, China; 2Heilongjiang Province Conservation Tillage Engineering Technology Research Center, Daqing 163319, China

**Keywords:** soybean seedling, root image, semantic segmentation, U-Net model, attention mechanism

## Abstract

Soybean seedling root morphology is important to genetic breeding. Root segmentation is a key technique for identifying root morphological characteristics. This paper proposed a semantic segmentation model of soybean seedling root images based on an improved U-Net network to address the problems of the over-segmentation phenomenon, unsmooth root edges and root disconnection, which are easily caused by background interference such as water stains and noise, as well as inconspicuous contrast in soybean seedling images. Soybean seedling root images in the hydroponic environment were collected for annotation and augmentation. A double attention mechanism was introduced in the downsampling process, and an Attention Gate mechanism was added in the skip connection part to enhance the weight of the root region and suppress the interference of background and noise. Then, the model prediction process was visually interpreted using feature maps and class activation mapping maps. The remaining background noise was removed by connected component analysis. The experimental results showed that the Accuracy, Precision, Recall, F1-Score and Intersection over Union of the model were 0.9962, 0.9883, 0.9794, 0.9837 and 0.9683, respectively. The processing time of an individual image was 0.153 s. A segmentation experiment on soybean root images was performed in the soil-culturing environment. The results showed that this proposed model could extract more complete detail information and had strong generalization ability. It can achieve accurate root segmentation in soybean seedlings and provide a theoretical basis and technical support for the quantitative evaluation of the root morphological characteristics in soybean seedlings.

## 1. Introduction

The root system is an important organ for water and nutrient uptake in soybean plants, and its morphological characteristics are closely related to the growth and development of above-ground organs, yield, quality and resistance [1]. There is considerable potential for breeding cultivars that can efficiently absorb water and nutrients by understanding the root morphological traits associated with soybean growth and development [2]. This makes the root important for identifying crucial traits in promising breeding targets [3]. However, the roots normally function within the soil, limiting direct observation [4]. Collecting root images from soil is time-consuming and laborious, and may damage root system architecture [5]. Fortunately, nutrient solution hydroponics is a cultivation method of growing plants in nutrient solution [6]. It solves non-visualization in soil cultivation, facilitates real-time observation and statistical information of root morphology, and is widely used in soybean seedling germplasm resource identification and root trait screening [7,8,9]. It was found that the seedling stage of the soybean has an important position in its entire reproductive period and is the best period for cultivating good root systems [10]. The quantitative study of root morphological characteristics during this period is of great significance for germplasm resource innovation, the selection and breeding of high-quality varieties, and root genetic improvement.

Root segmentation is a key technology in root morphological feature extraction and measurement methods. Segmentation results directly affect the subsequent measurement results of root morphological parameters. In the traditional root segmentation field, Liu et al. [11] proposed a rape root segmentation method based on color and Gaussian model, but it needs to set morphological parameters artificially and contains a small amount of noise in the background. The threshold-based segmentation method is sensitive to noise, but has a problem grouping targets with similar background and root color into one category, which is often unsatisfactory especially in the face of interference factors caused by noise such as water stains, noise spots and regions with insignificant contrast. She et al. [12] adopted an automatic global threshold segmentation method to segment cotton root images, which easily loses part of the roots in complex environments. Wang et al. [13] used the pixel classification background segmentation method based on a support vector machine to segment maize root images, but some disconnected regions and orphaned pixels would be generated. This segmentation method requires the manual extraction of features, which makes the algorithms more difficult. Falk et al. [14] proposed a soybean seedling root segmentation method based on a computer vision-imaging platform and machine learning. This platform delivers biologically relevant time-series data on root growth and development for phenomics and plant breeding applications.

In recent years, with the rapid development of convolutional neural networks [15], deep learning has become a research hotspot in the field of machine learning, showing the capabilities to address various challenging computer vision tasks. Instead of traditionally tedious manual target feature extraction, convolutional neural networks can automatically learn features from input data and achieve the end-to-end pixel-level classification of image targets. Long et al. [16] proposed fully convolutional networks (FCN), which introduced deep learning into the field of image semantic segmentation. The accuracy of semantic segmentation has been remarkably improved [17]. At present, more and more scholars have applied semantic segmentation algorithms based on deep learning to various fields such as medicine [18], semiconductor material [19], remote sensing [20] and agriculture [21,22,23], and have made great progress in plant root image segmentation. Wang et al. [24] proposed a fully automated soybean root segmentation method based on convolutional neural networks called SegRoot, validated its best segmentation performance by using a transfer learning technique and evaluated how different network capacities affected the performance. Teramoto et al. [25] adopted a U-shaped full convolutional neural network to semantically segment rice roots in trench profile images. Smith et al. [26] used a convolutional neural network U-Net to effectively segment chicory roots and soil in RGB root images. However, the feature layer of this traditional convolutional neural network assigns the same weight to the target features and interference features. Detailed information of the image will be gradually lost after multiple convolutional pooling operations, making root segmentation edge rough and root disconnection. Gong et al. [27] improved the U-Net model by introducing residual module and SENet (Squeeze-and-Excitation Networks) [28] to segment the root images of rice seedlings planted in transparent bags. Compared with the Otsu method, the proposed model can automatically segment the root morphology in rice root images under strong noise with higher accuracy. However, it also increases the complexity of the model. Kang et al. [29] proposed an attention mechanism-based semantic segmentation model for the in situ imaging of cotton root systems to distinguish the root system from the soil background. A simple and effective attention module, CBAM (Convolutional Block Attention Module) [30] is introduced into the model, so that the model has a good segmentation effect. However, the above image segmentation methods often obtain some false positive pixels when extracting roots from the original root image with water stains, noise spots and regions with insignificant contrast, causing partial root disconnection. It is necessary to further improve the network architecture.

This paper proposed a semantic segmentation model of soybean seedling root images based on an improved U-Net network for soybean seedling root images in a hydroponic environment. To achieve accurate soybean seedling root segmentation and meet the demand of fine root phenotype measurement, a U-Net-based network architecture was used. Inspired by the visual attention mechanism [31], the attention model was embedded in the downsampling and skip connection parts of the model, so that the model pays more attention to the root region and accurately identifies the feature of pseudo roots. To validate the rationality of the improved network, the model prediction process was visually interpreted using feature maps and class activation mapping maps. The connected component analysis method was used to remove the remaining noise in the prediction map. Furthermore, to further verify the generalization ability of the model, a segmentation experiment on soybean root images was performed in the soil-culturing environment. The segmentation network designed in this study can segment the root features of soybean seedlings more accurately in both hydroponic and soil-culturing environments, providing accurate and reliable data support for the measurement of soybean root parameters at the seedling stage.

## 2. Materials and Methods

### 2.1. Experimental Materials and Image Acquisition

Hydroponically grown soybean seedling roots planted in the research laboratory of the College of Engineering, Heilongjiang Bayi Agricultural University, were used as the experimental subjects. The experimental soybean variety was Beidou 37, provided by the Agricultural Extension Center of Jianshan Farm, Heilongjiang Province. Soybeans were planted in plastic cups containing half-strength Hoagland nutrient solution in a growing environment: daytime temperature (20 ± 3) °C and nighttime temperature (15 ± 3) °C. The cup bowl size was 90 mm in diameter at the top, 57 mm in diameter at the bottom, and 180 mm in height, and two seedlings were grown in each cup. The soybean seedling culture environment is shown in Figure 1. The image data required for the experiment were collected centrally on December 21–22, 2021, using a flatbed color image scanner with EPSON PERFECTION V800 PHOTO, and the images were saved in jpg file format, with the original resolution of each image being 2559 pixels × 4094 pixels. A total of 36 images of the root system at the seedling stage were collected.

### 2.2. Image Annotation and Data Augmentation

Using the Adobe Photoshop CC 2019 polygon lasso tool to manually annotate 15 randomly selected original soybean seedling root images from the above images, the soybean root area pixels in the root images were marked as 255, i.e., white, and the rest of the background pixels were marked as 0, i.e., black. Then, they were saved in png format as annotated images of the same size as the original images, and the image annotation is shown in Figure 2.

Considering the high resolution of the original root images, the computational complexity of the semantic segmentation algorithm and the consumption of GPU memory resources, the spatial dimension of the input images was set to 512 × 512 during model training. Since the input image size needs to match the network input size when training the network, the original images and their annotated images were cropped into sub-images of size 512 × 512 pixels to construct a soybean seedling root system image segmentation dataset. It was divided into training set, validation set and test set according to the ratio of 6:2:2, where the training set had a total of 240 images, the validation set had a total of 80 images and the test set had a total of 80 images. Detailed information about the total number of images, their size and their division for training, validation and testing stages is shown in Table 1.

The raw images were augmented to avoid network overfitting to increase the diversity of the dataset. In each iteration of training to read image data, a dynamic data augmentation method was used to randomly rotate the cropped image by 90° and 180°, adjust the hue saturation brightness (HSV), flip horizontally and vertically, add Gaussian noise and add salt and pepper noise. The response probability of each operation is 50%, as shown in Figure 3.

### 2.3. Conventional U-Net Network

The U-Net network [32] is a fully convolutional neural network obtained based on the optimization improvement of the FCN network. It has the advantage of small sample learning and can use fewer training samples for learning to achieve faster and more effective segmentation, mainly including encoder, skip connection and decoder. The encoder is located on the left side of the model with four sub-blocks. Each convolutional block contains two 3 × 3 convolutions, and then maxpooling is performed. The context information in the image is captured through the downsampling operation, so that the target features in the image can be extracted layer by layer. The decoder is located on the right side of the model and contains four sub-blocks. The upsampling operation is performed through deconvolution to restore the details of the object and the resolution of the feature map, so as to achieve accurate positioning. The deconvolution operation upsamples the feature map by using a 2 × 2 transpose convolution, which will double the size of the feature map and reduce the channel size of the feature map by half. With the help of two 3 × 3 convolutions and a 1 × 1 convolution, a feature map realizes the classification of each pixel, and generates a predicted segmentation map. The main feature of the U-Net network is the introduction of skip connection, which transmits the output from the encoder to the decoder. The feature fusion is realized by copying and cropping the feature map of these corresponding positions and the output of the upsampling operation in the channel dimension. The structure of the conventional U-Net network is shown in Figure 4.

## 3. Proposed Soybean Root System Image Segmentation Model

The structure of the improved U-Net network is shown in Figure 5. The detailed parameters of the proposed model are shown in Table 2. The main improvements introduced are: (1) The double attention mechanism is added in the process of network downsampling, which is located after the standard convolutional block consisting of two sets of 3 × 3 convolution and ReLU activation functions. The features updated by the double attention mechanism are passed layer by layer, so that the network can focus on the target features throughout the training process and solve the problem that the network relies on fixed weights in the process of image feature extraction. (2) The AG module is added to the skip connection part between downsampling and upsampling, which is located at the end of each skip connection. The module adjusts the output features of the encoder by receiving the feature map of the decoder at the next layer. After upsampling, the feature map is sent to the feature map of the encoder at the upper layer as a gate signal to activate spatial attention. The output features of the encoder are adjusted to further suppress the feature expression of irrelevant background regions, reduce the influence of noise on root segmentation and improve the root segmentation accuracy.

### 3.1. Dual Attention Mechanism

To achieve a focus on root system features, a dual-attention feature fusion module was used, including an External Attention (EA) [33] mechanism and Efficient Channel Attention (ECA) [34] mechanism in parallel form, as shown in Figure 6. The External Attention mechanism enhances the network’s ability to focus on the root site. The Channel Attention mechanism makes the network focus on the important feature channels. The EA and ECA mechanisms are used to construct a large range of semantic dependencies of features in the spatial dimension and channel dimension simultaneously. Then, the updated features in different dimensions are overlaid to capture important information in different dimensions to enhance the model representation and improve the root segmentation effect.

#### 3.1.1. External Attention Mechanism

An external attention mechanism was introduced to fully consider the relationship between each pixel point in the feature map to enhance the feature extraction capability in the spatial dimension. The structure of the EA module is shown in Figure 7.

A two-dimensional convolution with a convolution kernel size of 1 is used for the feature extraction of the input feature map to obtain a feature map of dimension C × H × W, which is reconstructed into a feature map of dimension C × N (N = H × W). This feature map is input to the first linear layer, i.e., the external attention mechanism $M_{k}$, for a one-dimensional convolution operation. Then, it is sent to the second linear layer, i.e., the external attention mechanism $M_{v}$, for a one-dimensional convolution operation after softmax activation processing and normalization. This feature map is reconstructed into a feature map of dimension C × H × W after a two-dimensional convolution with a convolution kernel size of 1. The feature map is element-wise summed with the input feature map, and the output feature map is obtained after the ReLU activation function. The specific equation is as follows.
(1)$$A=\mathrm{Norm}\left(FM_{k}^{T}\right)$$
(2)$$F_{out}=AM_{v}$$
where *M* is a learnable parameter independent of the input and acts as a memory unit for the whole training data set. Two different memory units $M_{k}$ and $M_{v}$ are used as keys and values, respectively. The input feature map *F* is matrix multiplied with the transpose of $M_{k}$ and then normalized to obtain the attentional feature map *A*. *A* is matrix multiplied with $M_{v}$ to obtain the output feature map $F_{out}$ processed by the attention mechanism.

The EA module is able to learn correlations between spatial locations during training, enabling the network to focus on root sites in the feature map, suppress irrelevant information such as background and enhance the detection rate of the network for lower-contrast root targets.

#### 3.1.2. Channel Attention Mechanism

Root system images taken under non-ideal conditions may contain extraneous noise. During the downsampling process of the network, this irrelevant information will be retained on some channels of the feature map, which will affect the segmentation accuracy of the root system. The channel attention mechanism was introduced to address the problems. The ECA module is improved based on SENet to enhance the important features and suppress the useless features by assigning the importance of information on the channel dimensions to improve the feature representation of the network model. The structure of the ECA module is shown in Figure 8.

A Global Average Pooling (GAP) operation is applied to each channel of the input feature map. Each two-dimensional feature map is compressed to a single real number with global perceptual field and then extracted to obtain a global feature map of dimension 1 × 1 × C. A fast one-dimensional convolution with a nondegenerate convolution kernel of size k is used instead of SENet fully connected. After the sigmoid activation function, the value is normalized to a range of 0 to 1, and the attention weight of each channel is generated. The attentional feature map is obtained by weighting each channel of the input feature map using this weight. *k* value is calculated as shown in the following equation.
(3)$$k=\frac{\log_{2}C+1}{2}$$
where *C* is the number of channels of the feature map.

The ECA module is able to learn the correlation between channels during the training process, suppress irrelevant noise interference such as water stains and noise, strengthen the feature weights of root regions and enhance the network’s ability to recognize root targets.

### 3.2. Attention Gate Mechanism

In some areas of root images, there is no obvious contrast between root and non-root at the pixel level. If the low-level feature map output by the encoder is directly connected with the high-level feature map generated by the decoder after upsampling, the deep detail information of the image will be lost. The shallow features that cannot eliminate the irrelevant noise in the skip connection will have an impact on the output results of the network model. Based on the direct connection, the Attention Gate mechanism [35] was added. The rich semantic information in the high-level feature map is used to select features for the low-level feature map, so that more detailed information can be added to the low-level feature map. The AG module can automatically learn to focus on target structures of different shapes and sizes, effectively suppress the expression of features in the target-irrelevant regions of the image and highlight the expression of features in target-related regions. The structure of the AG module is shown in Figure 9, where *g* is the feature map of the decoder after upsampling and x is the feature map of the encoder. The two feature maps are each subjected to a 1 × 1 convolution operation to make their channel numbers the same. Then, the features are summed to obtain the enhanced feature map. This feature map is processed by the ReLU activation function, and then a 1 × 1 convolution operation is performed on it. The attention weight coefficients are obtained by the action of the Sigmoid activation function. Resampler is used to resample the image to its original size. The resampled feature map is element-wise multiplied with the encoder’s feature map x to obtain the attention feature map.

By adding the AG module to the skip connection part of the network model, the feature response of the root region can be highlighted, the feature response of the background region can be suppressed, and the learning ability of the network for the root region can be improved.

## 4. Experiments and Results

### 4.1. Network Training

#### 4.1.1. Experimental Environment

Python 3.6 was used as the programming development language; the operating system environment was Windows 10 64-bit. The experimental hardware development environment was an Intel(R) Core (TM) i5-10400 CPU @ 2.9 GHz processor with 16 GB of running memory and an NVIDIA GeForce RTX 2060 graphics card. It was also equipped with CUDA 10.0 as the parallel computing framework and CUDNN 7.4.2 as the deep neural network acceleration library. The proposed model was implemented in the PyTorch 1.2.0 deep learning development framework.

#### 4.1.2. Loss Function

The selection of the loss function in the experiment directly affects the training results of the network. The loss function helps to optimize the parameters of the network, and the purpose of training a deep convolutional neural network is to find the optimal solution of the loss function. Since the semantic segmentation problem of soybean seedling root images is a binary classification problem with only two categories, root and background, the binary cross entropy (BCE) loss function is used. However, the number of root pixels in this dataset is a low proportion of the root images being detected, and the BCE loss causes the learning and recognition of the network toward the root category to be inhibited, and the network is more inclined to predict as background. Compared with BCE loss, Dice loss can solve the problem of positive and negative sample imbalance well [36]. Among them, the mathematical expressions of BCE loss and Dice loss are shown in Equations (4) and (5).
(4)$$Loss_{BCE}=-\frac{1}{N}\overset{N}{\underset{i=1}{\sum }}\left[y_{i}\log p_{i}+\left(1-y_{i}\right)\log\left(1-p_{i}\right)\right]$$
(5)$$Loss_{Dice}=1-\frac{2\sum_{i}^{N}y_{i}p_{i}+\varepsilon }{\sum_{i}^{N}y_{i}+\sum_{i}^{N}p_{i}+\varepsilon }$$
where N is the total number of pixel points in the root image; i is the *i*-th pixel point; $y_{i}$ is the label value of the *i*-th pixel point, which is defined as 1 if this pixel point is the root, and 0 otherwise; $p_{i}$ is the probability value of the *i*-th pixel point being predicted as the root, which takes values from 0 to 1. ε is a smaller positive value that exists to avoid the case of a denominator of 0, and its value is 1.

The combination of BCE loss function and Dice loss function is used as the loss function of the root segmentation model, which makes the model training stable and can effectively solve the problem of positive and negative sample imbalance [37], and the calculation formula is shown in the following equation.
(6)$$Loss=\lambda Loss_{BCE}+Loss_{Dice}$$
where Loss is the total loss; $Loss_{BCE}$ is the dichotomous cross-entropy loss, $Loss_{Dice}$ is the Dice loss, and λ is the weight coefficient to balance the importance between the BCE loss function and the Dice loss function.

The BCE loss judges the classification effect of each pixel point from details. The Dice loss judges the classification effect of the predicted segmentation map from the global perspective. Considering that the segmentation of root images needs to focus on the importance of global judgment, and the main task of the model is segmentation, the BCE loss weight is too large to make the model tend to predict as background, so *λ* is set to 0.3 in the design of the combined loss function.

#### 4.1.3. Experimental Parameter Setting

To ensure the repeatability of the model training experiment, a fixed random seed strategy was adopted. Each experiment produced the same input data to ensure that the model achieved the same results in each run. To avoid generating overlearning, the maximum number of training rounds was set to 10 epochs [38], and the batch size of each iteration of training was set to 1. The 240 training samples were iterated 240 times per round, for a total of 2400 iterations. The network model weights were randomly initialized using kaiming normal distribution strategy, and the training process used the Stochastic Gradient Descent (SGD) [39] algorithm based on momentum for network optimization and update, the initial learning rate was set to 0.01, the momentum factor was set to 0.99, and the weight decay factor was set to 0.00001 to prevent the network from overfitting. The dynamic learning rate adjustment strategy in the learning rate decay training strategy was used. After each epoch was trained, it was tested on the validation set and its F1 value was recorded. When it was detected that the F1 value of the validation set did not rise under 3 epochs, the learning rate was adjusted to 10% of the original one, and the training stopped after the number of iterative rounds reached the maximum.

### 4.2. Model Evaluation Metrics

To quantitatively evaluate the performance of the constructed network model in the semantic segmentation task of soybean root images, five commonly used measures of algorithm performance were used: Accuracy, Precision, Recall, F1-Score (F1) and Intersection over Union (IoU). The accuracy of the soybean root segmentation results was evaluated by five commonly used metrics of algorithm performance. These five metrics can be calculated by the confusion matrix [40]. The positive sample represents the root pixel and the negative sample represents the non-root pixel. The confusion matrix is shown in Table 3.

Accuracy refers to the proportion of the total number of correctly predicted pixels in the total number of all pixels. It is calculated as follows.
(7)$$\mathrm{Accuracy}=\frac{TP+TN}{TP+TN+FP+FN}$$

Precision refers to the proportion of the total number of pixels correctly predicted as root system to the total number of all pixels predicted as root system, highlighting the false detection rate in the prediction results. It is calculated as follows.
(8)$$\mathrm{Precision}=\frac{TP}{TP+FP}$$

Recall refers to the proportion of the total number of pixels correctly predicted as root systems to the total number of all pixels that are actually root systems, highlighting the rate of missed detections in the prediction results. It is calculated as follows.
(9)$$\mathrm{Recall}=\frac{TP}{TP+FN}$$

F1 is an evaluation metric that balances precision and recall and comprehensively reflects the segmentation effect, which is defined as the harmonic mean of precision and recall values in the range of (0,1). The larger the value, the better the segmentation effect of the model. Its calculation formula is as follows.
(10)$$F1=\frac{2\times \mathrm{Precision}\times \mathrm{Recall}}{\mathrm{Precision}+\mathrm{Recall}}$$

IoU refers to the ratio between the intersection and union of the prediction result of the root pixel category and the true label by the model. It is a metric to comprehensively evaluate the segmentation performance. The value of 1 indicates that the prediction result is completely consistent with the true label. The calculation formula is as follows.
(11)$$\mathrm{IoU}=\frac{TP}{TP+FP+FN}$$
where *TP* is true positive, which indicates the number of pixels that are manually labeled as root regions and automatically predicted as root regions by the model; *TN* is true negative, which indicates the number of pixels that are manually labeled as background regions and automatically predicted as background regions by the model; *FP* is false positive, which indicates the number of pixels that are *FP* is false positive, which indicates the number of pixels that are manually labeled as background region but automatically predicted as root region; *FN* is false negative, which indicates the number of pixels that are manually labeled as root region but automatically predicted as background region.

Since the model segmentation time is a more important metric in evaluating the performance of segmentation methods in practical segmentation applications, the single image segmentation time t is used as the evaluation criterion.

### 4.3. Model Performance Evaluation

The accuracy evaluation metric and loss value of the validation set are calculated after each epoch. Model segmentation performance metrics of the validation stage after 10 rounds of training are shown in Table 4. Precision and Recall of the improved U-Net model are 0.9895 and 0.9844, respectively. The comprehensive evaluation metrics F1 and IoU of the model segmentation performance are 0.9869 and 0.9742, respectively. The validation loss value is 0.0208, indicating that the model has high segmentation accuracy.

### 4.4. Comparison of Segmentation Methods

The Otsu method based on threshold segmentation (OTSU algorithm), the traditional SegNet [41] network model, the traditional PSPNet [42] network model, the traditional DeepLabv3+ [43] network model and the traditional U-Net network model were selected for comparison tests to further verify the effectiveness of the proposed method for soybean seedling image segmentation. Eighty images were randomly selected in the test set as input. The evaluation metrics were used to quantitatively analyze the segmentation methods. The average value of the evaluation metrics for each algorithm is shown in Table 5.

The segmentation performance of the improved U-Net model is better than the other five algorithms overall. There are fewer cases of mis-segmentation and over-segmentation. The single-image segmentation time of this model is 0.153s. The improved U-Net model has little difference with the traditional U-Net model, the traditional SegNet model, the traditional PSPNet model and the traditional DeepLabv3+ model in the single image segmentation time. The total time consumption of the improved U-Net model is larger than that of the other four traditional deep learning models. This is due to the introduction of the attention module and the existence of a certain time-consuming calculation of attention weights.

In order to compare the segmentation effects of the OTSU algorithm, traditional SegNet network model, traditional PSPNet network model, traditional DeepLabv3+ network model, traditional U-Net model and improved U-Net model more intuitively, the segmentation effects of the methods are visualized. Segmentation effect graphs of some images are shown in Figure 10.

There is a small amount of over-segmentation when using the OTSU algorithm to segment the root system. In other words, the background pixel points are classified as root pixel points. The OTSU algorithm can roughly segment the root region when the difference between foreground and background colors in the image is large and the texture is simple. However, for images with root colors similar to or the same as the background color, the segmented root region shows more cases of root disconnection, while the edges are not smooth enough. The deep learning model in general has a better segmentation effect than the traditional image segmentation algorithm. The root morphology has been clearly segmented with the traditional SegNet model, the traditional PSPNet model, the traditional DeepLabv3+ model and the traditional U-Net model. The taproot is correctly segmented, and part of the lateral roots is distinguished. However, these four traditional models still have a few roots over-segmented. Very few noise spots and water stains similar to roots are wrongly classified as roots. The edges of roots extracted by the traditional SegNet model and traditional PSPNet model are a little rough. There is adhesion between roots in the segmentation results of the traditional DeepLabv3+ model. The lateral roots with low color contrast cannot be effectively segmented by the traditional U-Net model. The improved U-Net model effectively solves the problem of a small number of disconnected root systems. The segmentation is more accurate and the details are better. It can accurately segment the disconnected lateral roots, but there are still a few stray points in the segmentation result.

### 4.5. Visualization of Feature Maps and Heat Maps

The feature map is helpful to understand the features learned by the network. The heat map can directly reflect the importance of a certain area in the image. The brighter the color, the more attention the network pays to this area. To better understand the improved U-Net model, the feature maps and heat maps were visualized in the last convolutional layer of the traditional U-Net model and the improved U-Net model, respectively, during the prediction process. The results shown in Figure 11 were obtained from the segmentation task of the soybean seedling root image.

From Figure 11a,b, it is observed that the traditional U-Net network has obvious defects in the learning effect on the local detail region of the target. It is shown that the target region of the root system is not identified and the segmented image shows a few stray points of noise. It is observed from Figure 11c,d that the improved U-Net network learns better and covers the main region of the target more comprehensively. In the fourth image, the roots in the lower left corner of the image framed by the red circle and red box were selected. As shown by the model locating the location of the target in the image, it can learn the root system features with lower contrast. The attention mechanism effectively integrates the information of low-dimensional semantic feature maps and high-dimensional semantic feature maps, and makes the improved network pay more attention to the area where the lateral roots are disconnected, which fully explains the rationality of the improved network.

### 4.6. Post-Processing Results and Analysis Based on Connected Component Area Threshold

In the segmentation results obtained by using the improved U-Net model, there still exist a few stray noise points, as shown in Figure 10e. The largest feature difference between the foreground target and the stray points is the area feature. Therefore, the area of the connected component was extracted for analysis to remove the stray points from the image. In order to extract the area feature parameter, the regions in the binary segmentation map with a gray value of 255 and isolated from each other were extracted and labeled separately using the connected component labeling method, as shown in Figure 12a. The number of pixels in the stray point area of an image is generally less than 50. Therefore, when analyzing each small area marked, the noise threshold is set to 50. That is to say, if the area of the connected area is less than 50 pixels, the noise is assigned to 0. The image result after removing the noise is shown in Figure 12b.

In Figure 12, each column consists of the connected component labeled images using different colors and the root system images after the removal of the clutter noise, respectively. By comparing Figure 12, from (a), it is observed that each isolated root region is labeled with different colors, which can more intuitively highlight the stray point noise; from (b), it is observed that the stray point noise with smaller areas is removed, leaving an accurate root segmentation result.

### 4.7. Results and Analysis of Root System Image Testing of Multiple Soybean Varieties at Seedling Stage

To further verify the generalization ability and practicality of the improved U-Net model, 30 images of each multi-variety soybean seedling root system collected from hydroponic and soil culture environments without model training were segmented. The model prediction and post-processing results of some images in hydroponic and soil culture environments, respectively, are shown in Figure 13 and Figure 14.

As can be seen in Figure 13b, the improved U-Net model does not show any significant performance degradation, except for the presence of very few stray points of noise in the prediction results. This result shows that the improved network has a good generalization ability to the root image of soybean seedlings in the hydroponic environment. It cannot only accurately segment single-species soybean seedling root systems, but also has a good segmentation effect in the multi-species soybean seedling root image segmentation task without model training. It can be seen from Figure 13c that the residual stray points in the model prediction results have been eliminated and accurate root segmentation results have been obtained, which meets the segmentation requirements needed for the study to a certain extent.

From Figure 14b, it can be seen that the improved U-Net model has no obvious performance degradation. However, there are a very few stray points of noise and lateral root end disconnection in the prediction results. It is verified that the improved network has strong generalization ability and better segmentation results in the root system image segmentation task of multi-species soybean seedlings in the soil culture environment without model training. From Figure 14c, it can be seen that the residual stray points in the model prediction results have been filtered out. More accurate root segmentation results have been obtained.

In summary, the segmentation method can handle the root system images of multiple soybean species at the seedling stage in both hydroponic and soil-culturing environments. The segmentation result has less background noise, strong generalization ability and practicality.

### 4.8. Discussion

The accurate segmentation of a root system from an image background is a prerequisite for the fine-grained measurement of root morphological characteristics. To effectively analyze the root image of soybean seedlings, advanced automatic image segmentation methods are required. More recently, deep learning methods can help automate this task. It is worth noting that the performance of deep learning-based segmentation methods partially depends on image quality. Considering that image quality is very important for root system research, we used a high-precision scanner to obtain high quality root images in this study. In future research, an effective image enhancement method, such as a generative adversarial network, will be explored for enhancing the quality of the root images, which can help improve the segmentation accuracy.

To illustrate the performance of the proposed model, we explored five deep learning-based image semantic segmentation models in the self-built soybean seedling root dataset. Comparing the segmentation results of all models, the improved U-Net model has a better segmentation effect than the other four traditional models. The PSPNet model uses the Pyramid Pooling Module (PPM) to provide global context information by different-region-based context aggregation. The U-Net model transfers complete feature map information (i.e., both location and pixel values) to the decoder in the channel dimension. However, the PSPNet model and U-Net model have difficulty segmenting the soybean seedling root system completely, and many root pixels are lost. More specifically, due to the low contrast of lateral root and background, most pixels of lateral roots are classified as background. The SegNet model transfers only max-pooling indices (i.e., location of feature maps) from encoder to decoder for feature concatenation and reconstruction. Based on DeepLabv3 with Atrous Spatial Pyramid Pooling (ASPP), the DeepLabv3+ model uses a simple and effective decoder module to capture sharper object boundaries. However, some small bright spots and water drops similar to the color of soybean roots are also segmented by the SegNet model and DeepLabv3+ model. There is still some residual noise in the mask. In addition, we also performed segmentation experiments on soybean seedling root systems in the soil culture environment without model training. Our experimental results on soybean seedling root images in hydroponic and soil culture environments demonstrated the excellent performance and high accuracy of the improved U-Net model for end-to-end fully automatic root segmentation. This is important for the study of soybean root phenotypic parameters.

However, the proposed method also has some limitations. For the soybean seedling root images under a soil culture environment, the improved U-Net model could not completely segment the end of the lateral roots when the contrast at the end of the lateral roots was relatively low. The main reason for this problem is that, when constructing the root semantic segmentation dataset, the sample size with an insignificant lateral root end contrast is relatively small. The root system in most images has high contrast with the background, and the proposed model is trained on the whole root system image in the hydroponic environment. In future research, the network model can be further improved to enhance the feature extraction ability of the model for the root system. In addition, the root segmentation dataset can be further improved, for example, by increasing the number of samples with insignificant lateral root contrast to enhance the ability of the model to extract lateral root features with low contrast.

## 5. Conclusions

In this paper, we proposed a root image semantic segmentation model based on an improved U-Net network with a dual attention and attention gate mechanism to segment soybean seedling roots automatically and precisely. The dual attention mechanism in the downsampling process automatically focuses on the root region and important feature channels of the image. The attention gate mechanism in the skip connection part can make full use of the spatial information in the encoder to bridge the semantic gap between the encoder and the decoder, and suppress the feature activation of irrelevant regions in the image.

The proposed model was trained, validated and tested with self-built soybean seedling root datasets. The experimental results on the root test dataset showed that the Accuracy, Precision, Recall, F1 and IoU of the improved U-Net model were 0.9962, 0.9883, 0.9794, 0.9837 and 0.9683, respectively. The segmentation time of a single image was 0.153 s. Compared with five other algorithms including the OTSU algorithm, traditional SegNet model, traditional PSPNet model, traditional DeepLabv3+ model and traditional U-Net model, the proposed model has the highest overall accuracy and the best segmentation effect. It can extract more accurate root edges and effectively solve the problem of a few broken roots. Through the visual experiment of feature maps and heat maps, it is verified that the proposed model is reasonable and better than the traditional U-Net model. The post-processing algorithm based on the area threshold of the connected component can remove the background noise present in segmentation and obtain accurate root segmentation. It provides a theoretical basis and technical support for the quantitative assessment of soybean seedling root morphological characteristics. In the future, we need to add a quantitative module of root phenotype parameters, so as to achieve complete soybean root phenotype analysis on the premise of ensuring accuracy.

## Figures and Tables

**Figure 1 sensors-22-08904-f001:**
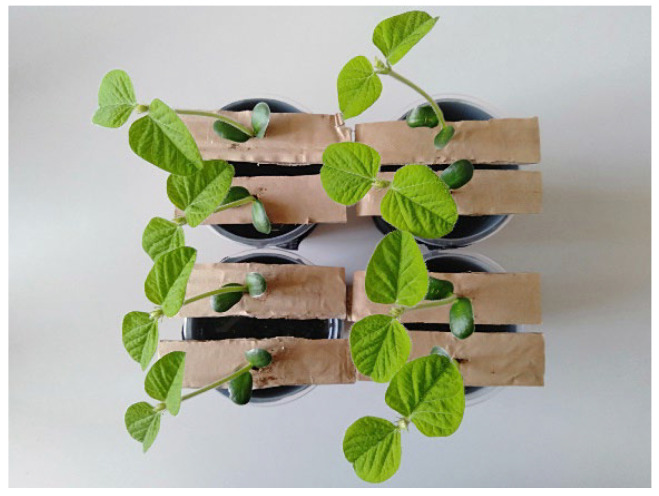
Soybean seedling culture environment.

**Figure 2 sensors-22-08904-f002:**
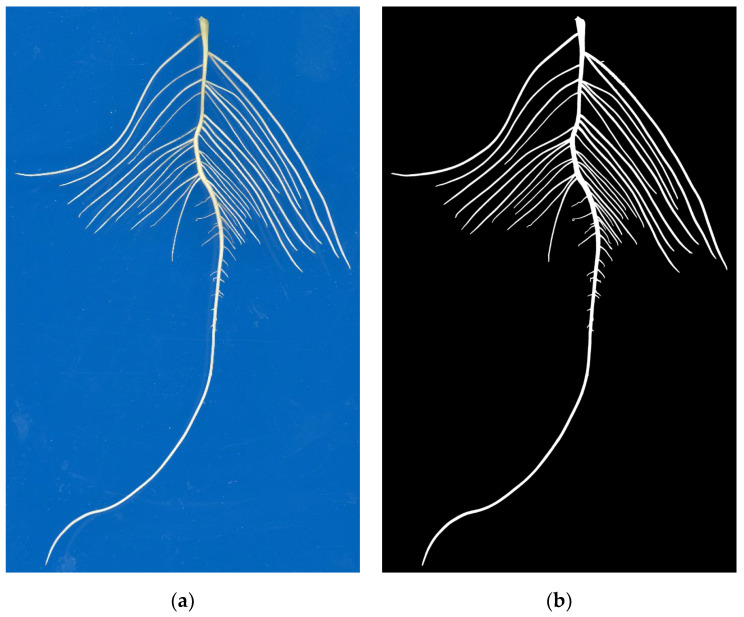
Image annotation of soybean root system. (**a**) Original image; (**b**) Annotated image.

**Figure 3 sensors-22-08904-f003:**
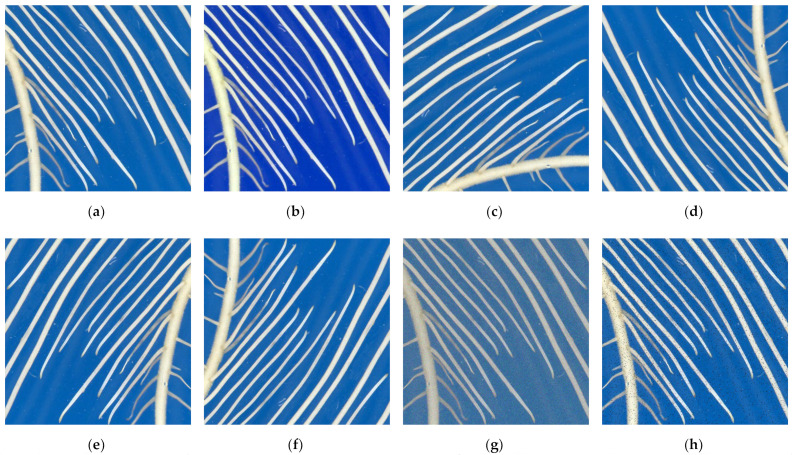
Root system picture after data augmentation. (**a**) Original image; (**b**) Adjust HSV; (**c**) Rotate 90° counterclockwise; (**d**) Rotate 180° counterclockwise; (**e**) Horizontal Flip; (**f**) Vertical Flip; (**g**) Gaussian noise; (**h**) Pepper noise.

**Figure 4 sensors-22-08904-f004:**
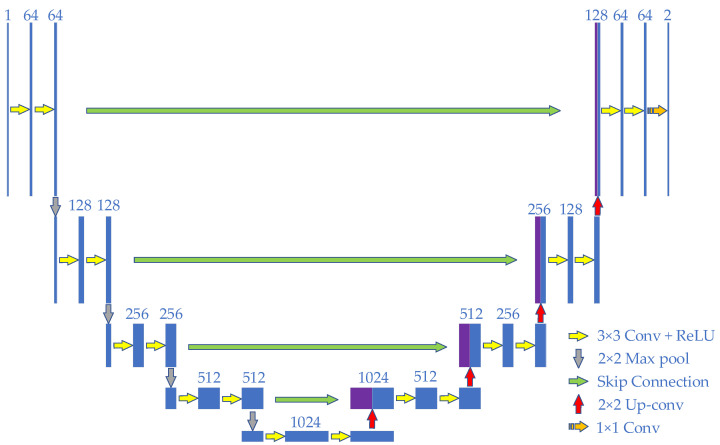
U-Net network structure diagram.

**Figure 5 sensors-22-08904-f005:**
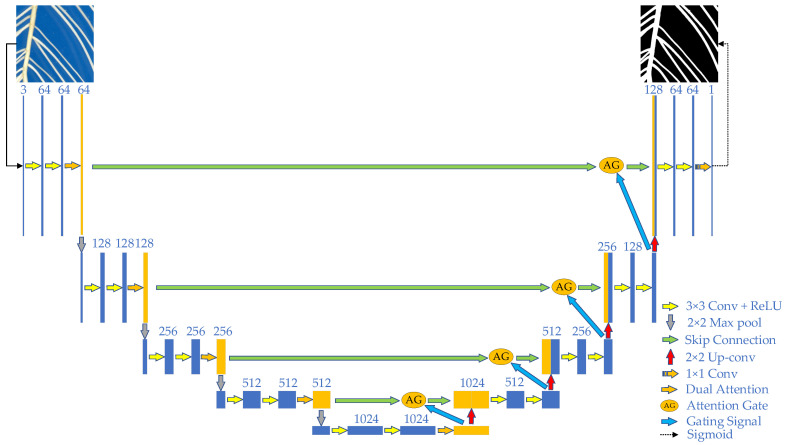
Structure diagram of semantic segmentation network of root images.

**Figure 6 sensors-22-08904-f006:**
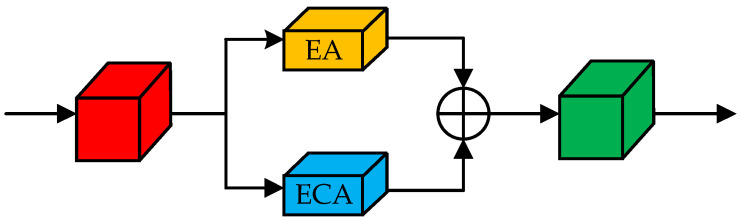
Dual attention module.

**Figure 7 sensors-22-08904-f007:**
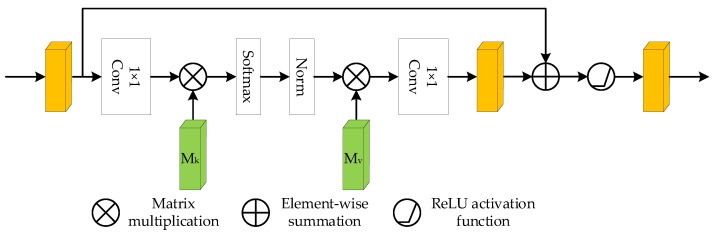
EA module.

**Figure 8 sensors-22-08904-f008:**
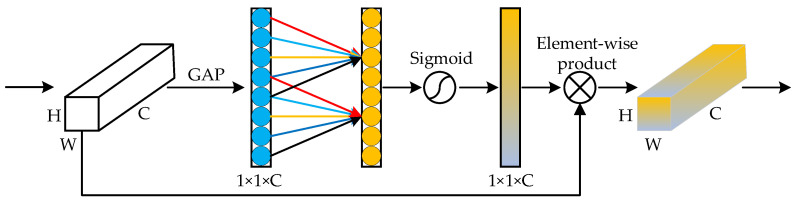
ECA module.

**Figure 9 sensors-22-08904-f009:**
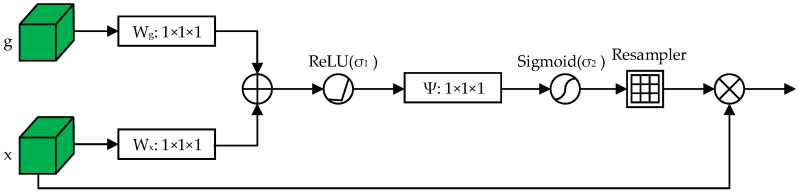
AG module.

**Figure 10 sensors-22-08904-f010:**
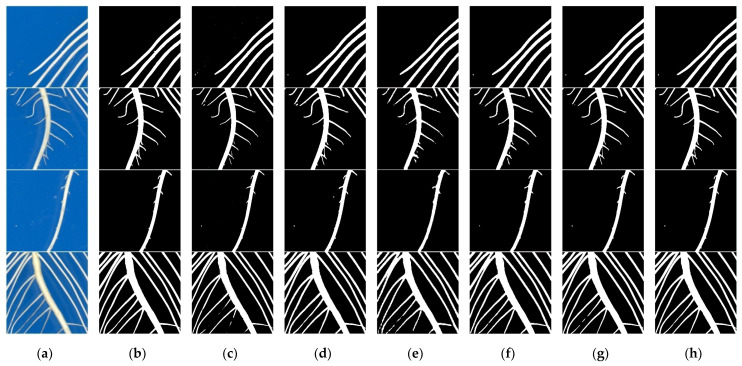
Comparison of segmentation results on different methods. (**a**) Original image; (**b**) Annotated image; (**c**) OTSU; (**d**) SegNet; (**e**) PSPNet; (**f**) DeepLabv3+; (**g**) U-Net; (**h**) Improved U-Net.

**Figure 11 sensors-22-08904-f011:**
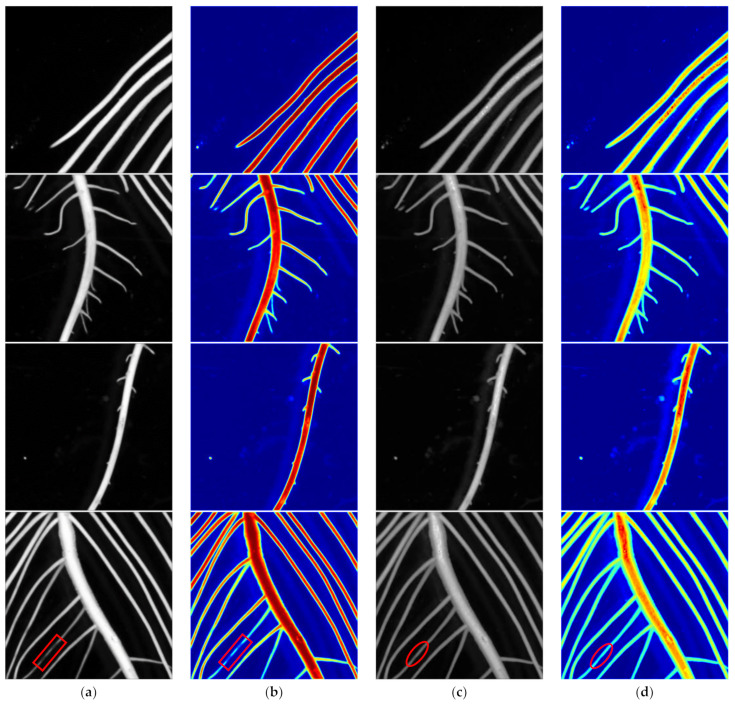
Visualization results of feature maps and heat maps on different models. (**a**) U-Net feature map; (**b**) U-Net heat map; (**c**) Improved U-Net feature map; (**d**) Improved U-Net heat map.

**Figure 12 sensors-22-08904-f012:**
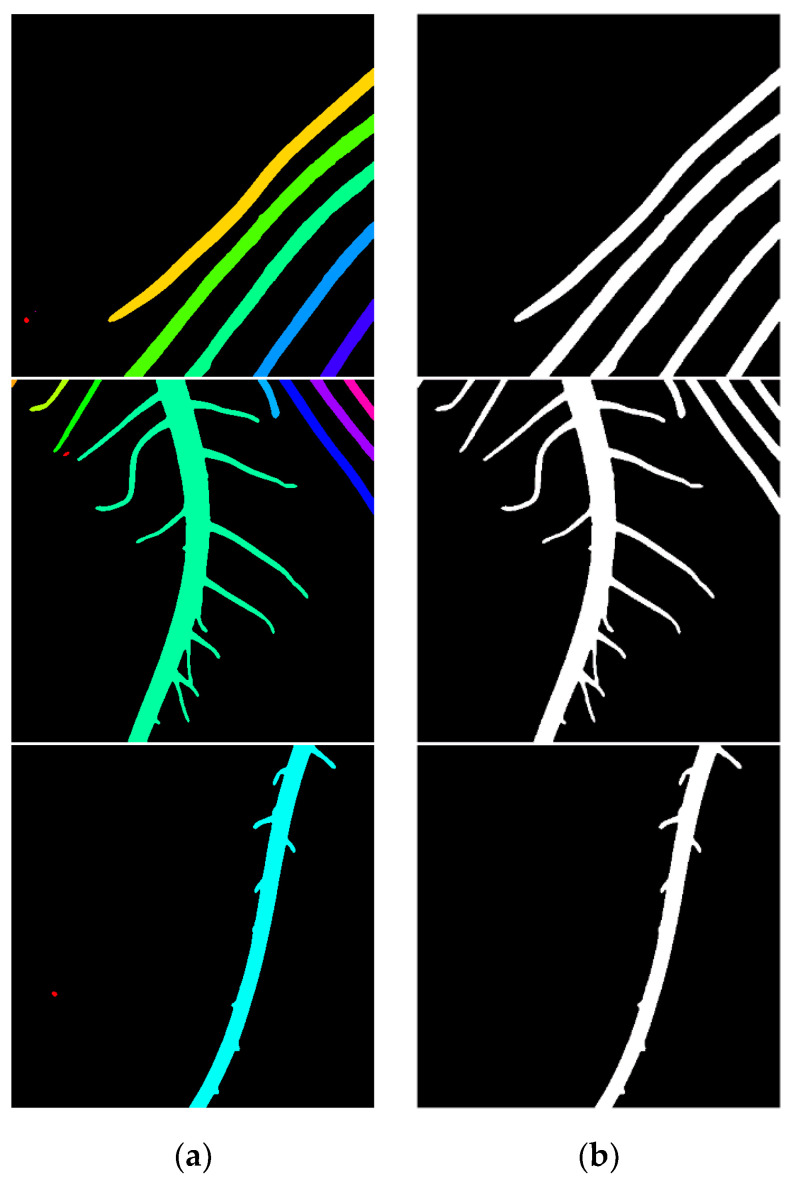
Connected component labeling and post-processing results. (**a**) Mark Connected component; (**b**) Image after stray noise removal.

**Figure 13 sensors-22-08904-f013:**
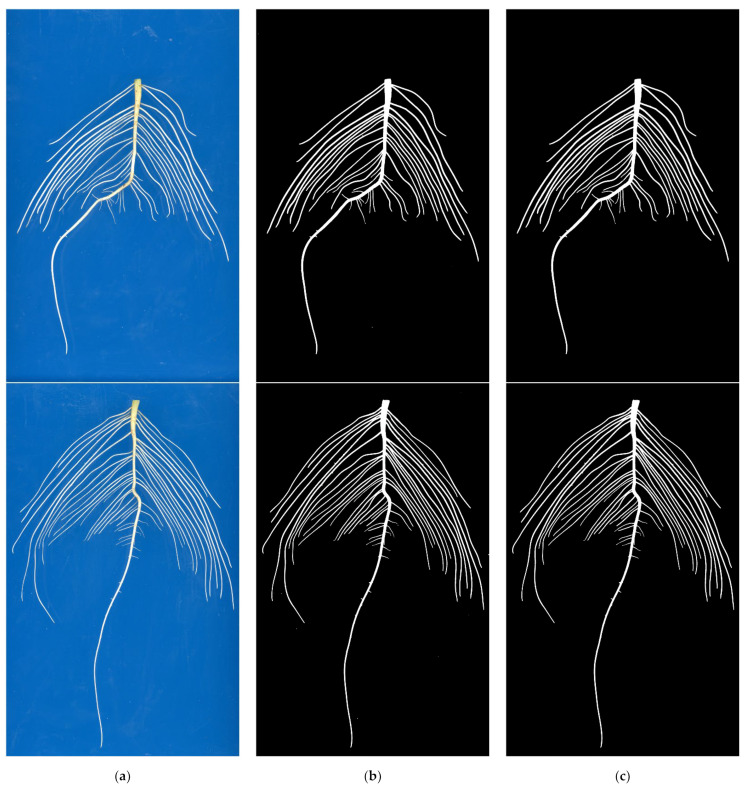
Image segmentation results of soybean seedling root system of multiple varieties under hydroponic environment. (**a**) Original image; (**b**) Model prediction result; (**c**) Post-processing result.

**Figure 14 sensors-22-08904-f014:**
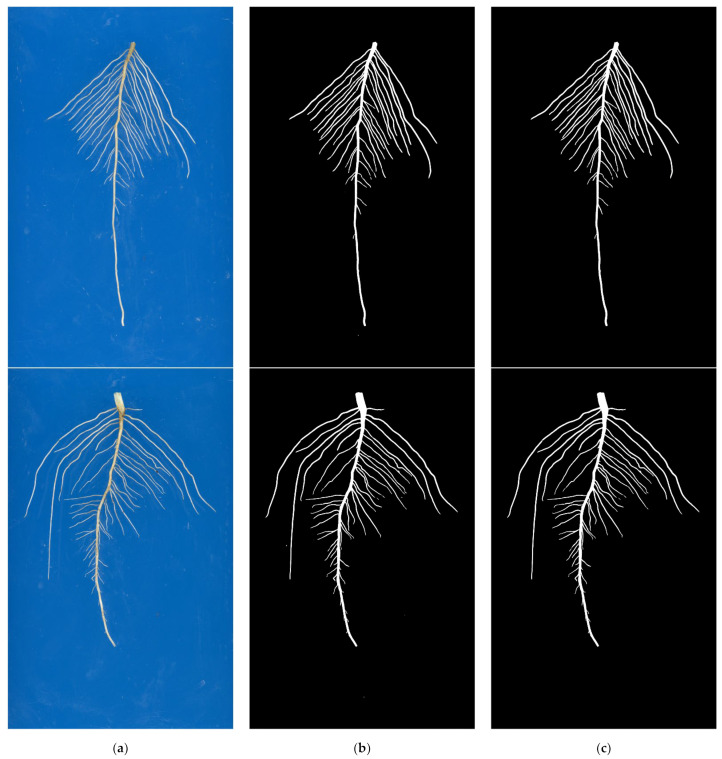
Image segmentation results of soybean seedling root system of multiple varieties under soil culture environment. (**a**) Original image; (**b**) Model prediction result; (**c**) Post-processing result.

**Table 1 sensors-22-08904-t001:** Detailed information about the total number of images, their size and their division for training, validation and testing stages.

Dataset Category	Image Size	Division Ratio	Sample Number
Training Set	512 × 512	60%	240
Validation Set		20%	80
Test Set		20%	80

**Table 2 sensors-22-08904-t002:** Detailed parameters of the proposed model.

Block	Type	Filter Size	Output Shape
Input Block			(512, 512, 3)
DownBlock1	Conv1	(3, 3)	(512, 512, 64)
	Dual Attention1		(512, 512, 64)
	Maxpool1	(2, 2)	(256, 256, 64)
DownBlock2	Conv2	(3, 3)	(256, 256, 128)
	Dual Attention2		(256, 256, 128)
	Maxpool2	(2, 2)	(128, 128, 128)
DownBlock3	Conv3	(3, 3)	(128, 128, 256)
	Dual Attention3		(128, 128, 256)
	Maxpool3	(2, 2)	(64, 64, 256)
DownBlock4	Conv4	(3, 3)	(64, 64, 512)
	Dual Attention4		(64, 64, 512)
	Maxpool4	(2, 2)	(32, 32, 512)
Middle Block	Conv5	(3, 3)	(32, 32, 1024)
	Dual Attention5		(32, 32, 1024)
UpBlock1	Up-conv1	(2, 2)	(64, 64, 512)
	AG1		(64, 64, 512)
	Conv6	(3, 3)	(64, 64, 512)
UpBlock2	Up-conv2	(2, 2)	(128, 128, 256)
	AG2		(128, 128, 256)
	Conv7	(3, 3)	(128, 128, 256)
UpBlock3	Up-conv3	(2, 2)	(256, 256, 128)
	AG3		(256, 256, 128)
	Conv8	(3, 3)	(256, 256, 128)
UpBlock4	Up-conv4	(2, 2)	(512, 512, 64)
	AG4		(512, 512, 64)
	Conv9	(3, 3)	(512, 512, 64)
Output Block	Conv1×1	(1, 1)	(512, 512, 1)
	Sigmoid		(512, 512, 1)

**Table 3 sensors-22-08904-t003:** Confusion Matrix.

Predicted Value	True Value	
	Root	Non-Root
**Root**	True Positive (TP)	False Positive (FP)
**Non-root**	False Negative (FN)	True Negative (TN)

**Table 4 sensors-22-08904-t004:** Improved U-Net segmentation performance evaluation on the validation set.

	Accuracy	Precision	Recall	F1	IoU	Loss
Improved U-Net	0.9964	0.9895	0.9844	0.9869	0.9742	0.0208

**Table 5 sensors-22-08904-t005:** Comparison of segmentation performance on different methods.

Segmentation Methods	Accuracy	Precision	Recall	F1	IoU	t/s
OTSU	0.9850	0.9801	0.8908	0.9298	0.8745	1.015
SegNet	0.9944	0.9691	0.9775	0.9726	0.9482	0.111
PSPNet	0.9846	0.9791	0.8934	0.9327	0.8749	0.133
DeepLabv3+	0.9941	0.9832	0.9647	0.9736	0.9492	0.112
U-Net	0.9951	0.9871	0.9726	0.9796	0.9606	0.118
Improved U-Net	0.9962	0.9883	0.9794	0.9837	0.9683	0.153

## Data Availability

Not applicable.

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
