# Peer review of "Soybean Seedling Root Segmentation Using Improved U-Net Network"

_sensors, 2022, doi:10.3390/s22228904_

Round 1
Reviewer 1 Report
This manuscript proposes a semantic segmentation model of soybean seedling root images based on an improved U-Net network, which introduces the double attention mechanism in the downsampling process and adds the Attention Gate mechanism in the skip connection part for soybean seedling root segmentation. In general, this research has certain significance for the application, but the following issues need to be addressed:
1.The introduction lacks logic and the relevance between paragraphs is not strong.
2. In Table 3, the manuscript only compares the proposed improved U-Net network with the other two methods, which is difficult to reflect the performance advantages of the model.
3. It is better to put "2.3. Conventional U-Net Network" in "3. Proposed Soybean Root System Image Segmentation Model".
4.In Figure 13, only a kind of soybean seedling is shown, which is contradictory with "It can not only accurately segment single specifications soybean seeding root systems, but also has a good segmentation effect in the multi specifications soybean seeding root images segmentation task without model training." in line 444. In addition, the same problem exists in 460 line of the manuscript.
5. In Figure 11, 13, and 14, the comparison of soybean seedling root image segmentation before and after the model improvement is not obvious, which is inconsistent with the statement in the text.
6.The depth of discussion of "4.8. Discussion" in Section 4.8 is insufficient.
7.There exist some other issues in this manuscript. For example, the text in Figure 4 and Figure 5 is too small; the title "Dataset Production and Data Augmentation" of Section 2.2 is improperly enunciated.
Reviewer 2 Report
The paper can be accepted after revisions:
1- Keywords: Do not use the words in the title of the article as keywords.
2- Introduction suffers from lack of motivations and innovations. It should be expanded to include a more detailed discussion of current problems. What is the importance and application of machine learning in the agriculture and food industry? In the introduction, it is better to discuss the importance of using deep learning and the Convolutional Neural Network. The purpose of the research is not well reported in the final part of the introduction. The motivation of the paper is not sufficiently justified in the introduction.
3- A section on network training should be included which details on the parameters used for training CNNs (what filter size?). what is the architecture of CNN? Such information would increase the quality of manuscript and also provides a way for re-implementation by others.
4- Please compare your results and research with the results of other authors and underline what is the scientific novelty in this work.
5- Provide information about the total number of images, their size and their division for training, validation and testing stages in a table.
6- The interpretation of results should be well written in the paper. The results need further technical discussions.
7- The Limitations of the proposed study need to be discussed before conclusion.
8- The findings in "Conclusions" Section should be stated point by point. A contextualization has to be added as incipit, in order to make the Conclusions section self-standing. Please make a few-line conclusion about the work. It can be the essence of all the results. It should be suggestive about the best practice for future works.
Reviewer 3 Report
This manuscript presented a semantic segmentation model of soybean seedling root images based on an improved U-Net network to address the non-structural problems, enabling to extract more complete detail root information, and had strong generalization ability. In addition, several aspects, including language expressing, advantageous test results compared to the results resulted from existing segmentation methods, and comprehensive discussion, etc., are presented well. However, I have some concerns with the manuscript as follows:
(1) Are the data and code produced freely available (which I encourage the authors to propose, to allow full replication of their experiments, and to increase the scientific impact of their work)?
(2) L64, check the “a U-based full convolutional neural network”.
(3) The word “Where” doesn't need to be indented in the first line and should appear in lowercase. Check L197, 225, 279, 288, 339 and so on.
All in all, this work is interesting, well organized and in the scope of the journal. I think that some additional information and clarification (mentioned before), should contribute to increase the impact of this work, which is in my opinion of good quality and is developed in the right direction.
Reviewer 4 Report
Aiming at the problems of over segmentation, unsmooth edge segmentation and root separation of soybean seedling images due to background interference, an improved UNet segmentation network was proposed. It is reasonable to improve the performance of model segmentation by adding attention mechanism, but there are still some problems to be further explained.
1. The introduction does not cover the relevant literature, which needs to be further supplemented.
2. The quality of your data is low, and the conclusion does not highlight the key novelty of the research.
3. There are few experimental contents and no comparison with some current advanced related methods
4. It is not comparable with the relevant research methods in the same type of literature, and fails to highlight the progressiveness of this research.
5. The English level of the manuscript needs to be further improved.
Round 2
Reviewer 1 Report
The manuscript has been revised according to the review comments, but there are still the following problems:
1. The introduction still lacks logicality, and the relevance between paragraphs is not strong. Each paragraph is just a pile of references without the analysis and summary. The above issues had been mentioned in the first round review, but the revision is limited.
2. The manuscript only compares the proposed improved U-Net network with the other three methods. It is suggested to make a comparative analysis with the current mainstream image segmentation models such as DeepLab v3 and PSPNet. The above issues had been mentioned in the first round review, but the revision is limited.
3. In Figure 11, the comparison of soybean seedling root image segmentation before and after the model improvement is not obvious, which is inconsistent with the statement in the manuscript. The above issues had been mentioned in the first round review, but the revision is limited, and there is still no obvious difference. It is suggested to give a mark or a partial diagram.
4. The depth of discussion is insufficient. The above problems have been mentioned in the first round of reviews, but this problem has not still been solved.
Reviewer 4 Report
For the revised manuscript, there are still some problems that need further explanation.
1. Further enrich the relevant literature review in the introduction.
2. In the contrast experiment with other segmentation methods in section 4.4, only one SegNet model was added, which failed to highlight the progressiveness of the improved UNet model proposed in this paper.
